# Impact of Dietary Sodium Butyrate and Salinomycin on Performance and Intestinal Microbiota in a Broiler Gut Leakage Model

**DOI:** 10.3390/ani12010111

**Published:** 2022-01-04

**Authors:** Mohammad Naghizadeh, Laura Klaver, Anna A. Schönherz, Sundas Rani, Tina Sørensen Dalgaard, Ricarda Margarete Engberg

**Affiliations:** 1Department of Animal Science, Aarhus University, Blichers Allé 20, 8830 Tjele, Denmark; lauraklaver96@gmail.com (L.K.); anna.schonherz@anis.au.dk (A.A.S.); sundas.rani@iiu.edu.pk (S.R.); tina.dalgaard@anis.au.dk (T.S.D.); 2SA-Center for Interdisciplinary Research in Basic Sciences, Faculty of Basic and Applied Sciences, International Islamic University, Islamabad 44000, Pakistan

**Keywords:** gut leakage, dysbacteriosis, 16S rRNA gene, amplicon sequencing, gut health, butyrate, and salinomycin

## Abstract

**Simple Summary:**

Imbalanced microbiota related to intestinal leakage is an emerging health problem in commercial broilers associated with partial loss of gut function and reduction in growth performance. Salinomycin is an ionophore coccidiostat with antibiotic effect, which is used as additive in broiler diets to control enteric diseases and improve performance. Due to the growing concern of the development of bacterial antibiotic resistance, ionophores are expected to be banned, within the EU, as feed additives in the near future. Butyrate with multiple beneficial effects on growth performance and pathogen control in broilers has been introduced as a promising alternative. In this study, the ability of a coated butyrate product to alleviate intestinal imbalance was compared to that of salinomycin after enteric challenge. Compared to butyrate and non-supplemented control, salinomycin increased potentially beneficial *Ruminococcaceae* and reduced potentially pathogenic *Enterobacteriaceae*, and counts of *Lactobacillus salivarius* and *Clostridium perfringens.* Further, salinomycin improved broiler performance. Dietary inclusion of coated butyrate had only limited effects on the composition of the broiler microbiota. Whether improved growth and feed utilization in the salinomycin-supplemented animals can be explained by suppression of *C.* *perfringens* and *L.* *salivarius* and enrichment of butyrate- and lactic acid-producing bacteria (*Ruminococcaceae* and *Lactobacillaceae*) needs further verification.

**Abstract:**

Unfavorable alterations of the commensal gut microbiota and dysbacteriosis is a major health problem in the poultry industry. Understanding how dietary intervention alters the microbial ecology of broiler chickens is important for prevention strategies. A trial was conducted with 672 Ross 308 day-old male broilers fed a basic diet (no additives, control) or the basic diet supplemented with 500 mg/kg encapsulated butyrate or 68 mg/kg salinomycin. Enteric challenge was induced by inclusion of 50 g/kg rye in a grower diet and oral gavage of a 10 times overdose of a vaccine against coccidiosis. Compared to control and butyrate-supplemented birds, salinomycin supplementation alleviated growth depression. Compared to butyrate and non-supplemented control, salinomycin increased potentially beneficial *Ruminococcaceae* and reduced potentially pathogenic *Enterobacteriaceae* and counts of *Lactobacillus salivarius* and *Clostridium perfringens*. Further, salinomycin supplementation was accompanied by a pH decrease and succinic acid increase in ceca, while coated butyrate (0.5 g/kg) showed no or limited effects. Salinomycin alleviated growth depression and maintained intestinal homeostasis in the challenged broilers, while butyrate in the tested concentration showed limited effects. Thus, further investigations are required to identify optimal dietary inclusion rates for butyrate used as alternative to ionophore coccidiostats in broiler production.

## 1. Introduction

The gastrointestinal (GI) tract of broilers is colonized by a tremendous number of microorganisms, collectively termed the microbiota [1]. Recent advances in microbiome research revealed that the gut microbiota composition and its products have an irreplaceable health impact, through a range of physiological functions, including supporting resistance to pathogens, affecting the immune system [2], controlling epithelial cell proliferation and differentiation [3], contributing to host nutrition and metabolism [4], regulating glucose metabolism and insulin resistance [5], and affecting the behavioral and neurological functions of the host [6].

Dysbacteriosis typically occurs around 3 weeks of life in commercial broiler chickens and produces a cascade of reactions in the GI tract, including gut inflammation, barrier dysfunction, and increased risk of bacterial translocation [7]. Dysbacteriosis has been associated with partial loss of GI tract function and reduction in growth performance and litter quality [8]. Besides direct impact on the host intestinal and immune cells, dysbacteriosis impairs intestinal integrity and increases intestinal permeability indirectly via changing the host intestinal homeostasis and fermentation metabolites [7], such as short-chain fatty acids (SCFAs) [9]. This condition in fact allows antigens and immune stimulants, such as pathogenic bacteria or their toxins, to penetrate internal organs and trigger systemic inflammation [8,10].

Salinomycin, an ionophore coccidiostat, is widely used as supplement in poultry feed to control infection with *Eimeria* spp. and to inhibit *Clostridium perfringens*-induced necrotic enteritis in poultry [11]. Due to the growing concern of the development of bacterial antibiotic resistance [12], ionophores are expected to be banned as feed additives in the near future [13]. A withdrawal of these compounds will most likely lead to an increase in cases of enteric diseases and performance problems in broiler chickens. To overcome the increased rate of enteric disorders in broilers, different dietary interventions have been proposed [14]; among these, butyrate and its derivatives have gained increasing attention. This is due to multiple potentially beneficial effects of butyrate on the epithelial barrier integrity via regulation of the assembly of tight junctions [15], epithelial cell proliferation [16], and reduction of apoptosis in the crypt compartment [17]. In addition to the role of butyrate in growth performance and pathogen control, anti-inflammatory properties of butyrate in chicken have been documented (reviewed in Reference [17]).

Butyrate is corrosive and volatile in nature; therefore, various forms of butyrate, including uncoated and enteric coated forms protected with fat or fatty acid salts, have been developed to make it easy to handle and less odorous and to prevent its dissociation in a bird’s proximal digestive tract [17].

Due to a vague definition of dysbacteriosis, an optimal experimental model for dysbacteriosis does not exist. Different approaches, therefore, have been used so far to induce gut barrier failure and inflammation [10,18,19,20]. In two studies by Tellez et al. [19] and Baxter et al. [20], gut barrier failure was induced in broilers involving increased intestinal digesta viscosity via inclusion of an extraordinary rye rich diet (58%). Chen et al. used a rye (34%), wheat (20%), and barley (10%) grower diet combined with a coccidiosis vaccine (2× overdose) to induce inflammation and gut barrier failure [10]. In the current study, gut inflammation and gut barrier failure were induced by inclusion of 5% rye in the diet and a 10× overdose of a commercial coccidiosis vaccine, as previously described by Naghizadeh et al. [21] and Engberg et al. [22]. In the latter model, birds supplemented with salinomycin showed reduced fluorescein isothiocyanate-dextran (FITC-D) and D-lactate plasma levels (indicators of gut leakage), reduced bacterial translocation to the liver and upregulated expression of tumor necrosis factor α (TNFα), and mucin (MUC2) genes in the intestinal tissue. It is assumed that the beneficial health effect of salinomycin in chickens is linked to changes in gut microbial composition, as well as physiological homeostasis. However, the differential regulatory effects of butyrate and salinomycin on gut microbiota composition have remained unclear. The primary objective of this study was to evaluate the effect of dietary butyrate and salinomycin supplementation on performance of broiler chickens under experimental enteric challenge conditions. Further, the aim was to investigate potential beneficial effects of dietary butyrate in comparison with salinomycin on composition and activity of the intestinal gut microbiota.

## 2. Material and Methods

### 2.1. Experimental Design

A total of 672 day-old male broiler chickens (Ross 308) were included in the study. The birds were wing-tagged and reared in pens with a floor area of 1.7 m^2^ covered with a thin layer of wood shavings as bedding material, with automatic control of temperature, light, and humidity. The study was conducted with 3 experimental groups in 24 pens with 28 chickens/pen, resulting in 8 replicates per treatment. Broilers were fed diets containing either salinomycin, Sacox 120^®^, (Huvepharma, Antwerp, Belgium), 69 mg/kg feed, or the butyrate product UltraGuard™-DUO, 0.5g/kg feed (Devenish, Belfast, Northern Ireland), or no dietary supplement (control) from day one and onwards. The supplements were used in concentrations recommended by the companies. An enteric challenge model was applied, where all birds were fed a wheat-based grower diet containing 5% rye from day 10 and were orally challenged with a 10 times overdose of an attenuated live vaccine against coccidiosis (Hipracox^®^-Hipra, Amer (Girona, Spain) on day 17. All chickens received a pelleted compound starter feed from day 1–10 and a pelleted compound grower feed from day 11–35 (Appendix A), and both feed mixtures were formulated without enzymes. Fresh feed and water were provided ad libitum throughout the experiment. Body weight, drinking water, and feed intake of the chickens in each pen were recorded weekly, and feed conversion (FCR) and water:feed ratios were calculated. The drinking water was offered in containers, which were weighed before and after filling, allowing the calculation of the content weight. The body weight of dead birds was recorded daily and was considered when calculating FCR.

### 2.2. Sample and Data Collection

At three different time points (day 21, 24, and 29 of age), 32 broilers from each dietary treatment group (four birds per pen) were randomly selected, weighed, and euthanized by means of CO_2_. The chickens were immediately dissected to obtain samples of intestinal contents. The content of the ileum, from Meckel’s diverticulum to the ileocecal junction, and both ceca of 4 broilers per pen were collected and pooled by segment before further analyses. The pooled samples were thoroughly mixed, and pH was measured using a Mettler Toledo pH meter combined with a glass reference electrode LE410 (Mettler Toledo GmbH, Greifensee, Swizerland). Sub-samples were taken for the enumeration of selected bacteria groups by culture. The remaining ileum and ceca samples were snap-frozen in liquid nitrogen for subsequent measurement of SCFA, lactic acid concentrations, and 16S rRNA gene amplicon sequencing.

### 2.3. Bacterial Enumeration

Plate count enumeration of selected groups of intestinal bacteria was performed at each of the three time points as described previously [23]. In brief, 1 g of each pooled intestinal digesta sample was rapidly transferred under a flow of CO_2_ into flasks containing 90 mL of a pre-reduced salt medium. This suspension was poured into CO_2_-flushed plastic bags and homogenized for 2 min using a stomacher laboratory blender (Interscience, St. Nom, France). Subsequently, plating on different media was performed after 10-fold serial dilutions. Lactic acid bacteria were counted on Man Rogosa and Sharp (MRS) agar (VWR, Leuven, Belgium) after anaerobic incubation at 37 °C for 48 h. *Lactobacillus salivarius* was identified by typical colony morphology on MRS agar and counted separately [23]. *C. perfringens* was enumerated on Tryptose Sulphite Cycloserine (TSC) agar (VWR, Belgium) after anaerobic incubation at 37 °C for 24 h. Coliform bacteria and lactose-negative enterobacteria were counted as red and white colonies, respectively, on MacConkey agar (VWR, Belgium) after aerobic incubation at 37 °C for 24 h. Enterococci were counted on Slanetz and Bartley agar (Merck, Darmstadt, Germany) after aerobic incubation at 37 °C for 48 h. Following logarithmic transformation, bacterial numbers were expressed as log colony forming units (CFUs) per gram of digesta.

### 2.4. Analysis of Short-Chain Fatty Acid and Lactic Acid

The concentrations of short-chain fatty acid (SCFA) and lactic acid in digesta samples were measured as previously described [24]. Briefly, intestinal digesta (10 g), 10-fold diluted with a 0.028 M sodium hydroxide solution, were extracted using 0.5 mL of concentrated HCl and 2 mL of diethyl ether, centrifuged (3000× *g* for 10 min at 4 °C), and derivatized with 10 μL of the N-methyl-N-t-butyldimethylsilyl-trifluoroacetamide (Merck Life Science A/S, Søborg, Denmark) after incubation at 80 °C for 20 min, and at room temperature for 48 h. Quantification of SCFA, and lactic acid, was performed on a Hewlett Packard gas chromatograph (Model 6890, Hewlett Packard, Agilent Technologies, Naerum, Denmark) configured with flame-ionization detectors and a capillary column. All samples were analysed using 2-ethylbutyric acid (Merck Life Science A/S, Søborg, Denmark) as an internal standard.

### 2.5. DNA Extraction

DNA was extracted from intestinal samples using an E.Z.N.A.^®^ Stool DNA Kit (Omega Bio-tek, Inc.; Norcross, GA, USA) according to the manufacturer’s instructions, applying the following modifications: 700 µL of lysis buffer SL2 provided in the kit was added to 250 mg of intestinal digesta. Samples were disrupted in a FastPrep-24TM benchtop homogenizer using metal beads instead of Glass Beads X, to enhance obtained DNA concentration. DNA was eluted in a final volume of 50 µL of elution buffer, DNA purity (OD260/OD280 ratio of ~1.8) was evaluated with the Nanodrop ND1000 (Thermo Scientific, Waltham, MA, USA), and quantified fluorometrically with Qubit 3.0 HS dsDNA assay (Life Technologies, Thermo Fisher Scientific, Waltham, MA, USA ). DNA concentrations were adjusted to 5 ng/μL using DNase I, RNase-free water (Life Technologies, Thermo Fisher Scientific, Waltham, MA, USA ), instead of the elution buffer, as the elution buffer had an inhibitory effect on the PCR.

### 2.6. 16S rRNA Gene Amplicon Sequencing

Amplicon libraries were prepared as described by the Illumina protocol [25]. Briefly, a PCR of 25 cycles was performed to amplify the V3–V4 region of the 16S rRNA gene prior to the incorporation of the Illumina overhang adapters (Second PCR: 10 cycles) using 2.5 µL of DNA sample (5 ng/µL), 0.5 µL of each primer, and 12.5 µL of 2x KAPA HiFi Hotstart ReadyMix (Kapa Biosystems, Merck, Darmstadt, Germany). Bacterial communities were assessed using universal primers (Bac341F: CCT ACG GGN GGC WGC AG and Bac805R: GAC TAC HVG GGT ATC TAA TCC) [26] covering the V3–V4 regions of the 16S rRNA gene. The third PCR amplification (8 cycles) was carried out using Nextera XT index primers (Illumina, San Diego, CA, USA). Amplifications were conducted on a Veriti^®^ 96-Well Thermal Cycler (Applied Biosystems^®,^ Thermo Scientific, Waltham, MA, USA). The PCR products were purified with AMPure XP beads and Veriti^®^ 96-Well Thermal Cycler (Applied Biosystems^®,^ Thermo Scientific, Waltham, MA, USA), as described in the protocol. Amplicon libraries were then sequenced on the Illumina MiSeq (Illumina, San Diego, CA, USA), generating 2 × 300 bp paired-end reads.

### 2.7. Statistical Analyses

#### 2.7.1. Statistical Analysis of Performance Data

Statistical analysis was performed using the statistical software R 3.2.3 in R studio v3.6.2 [27]. Performance data (body weight gain, feed intake, feed conversion ratio, and water/feed ratio) were compared among dietary treatment groups using a one-way ANOVA followed by Tukey’s multiple comparison test. The impact of dietary treatments and age on bacterial counts and concentrations of organic acids were analyzed using a linear mixed model, using the *lmer* function in the R package *lme4* [28]. Dietary treatment, age, and interaction between dietary treatment and age were included as fixed effects, while pen was included as random effect. The fixed effects were tested using an F-test with Kenward-Roger approximation, where the reduced model was tested against the full model by likelihood ratio tests using the *anova* function. When a fixed effect was found to be significant, a post-hoc test was performed, using the R package *multcomp* [29], and Tukey adjustment to correct for multiple comparisons. *p*-values ≤ 0.05 were considered statistically significant.

#### 2.7.2. 16S rRNA Sequence Read Processing and Quality Control

The demultiplexed sequence data of FASTQ format were processed with QIIME2 v. 2019.10 [30]. To infer amplicon sequence variants (ASVs), reads were quality filtered, trimmed, denoised, merged, and dereplicated, and PCR chimeras were removed using the DADA2 algorithm [31] implemented in Qiime2. Briefly, primers were trimmed removing 20 bases from the beginning of the forward and 23 bases from the beginning of the reverse read (removing bases corresponding to primer length), and bases of poor quality were removed truncating forward and reverse reads at 280 bp and 265 bp, respectively. DADA2 default parameter settings were applied otherwise. Detected ASVs were taxonomically classified using a V3-V4 specific Naive Bayes classifier [32]. The classifier was trained on 99% similarity clustered 16S rRNA gene sequences extracted from the SILVA v132 reference database [33] and trimmed to only include the V3-V4 region bound by the Bac341F/ Bac805R primer pair. Finally, the phylogeny was inferred using the MAFFT multiple sequence alignment method [34] and the FastTree algorithm [35] of the q2-phylogeny plugin implemented in QIIME2 with default settings.

#### 2.7.3. Statistical Analysis of 16S rRNA Sequencing Data

Exploratory microbial data analyses were performed with R version 3.6.2 [27] using the packages *Phyloseq* v1.30.0 [36], *vegan* v2.5-6 [37], *microbiome* v1.8.0 [38], *DESeq2* v1.26.0 [39], *stats* v3.6.2 [27], *ampvis2* [40], and *ggplot2* v3.3.1 [41].

Prior to data analysis, ASVs of unknown taxonomy at phylum and class level (22 ASVs out of 2686), samples with read depth below 1000 (1 sample), outliers based on preliminary ordination plot exploration (2 samples), and taxa known to be non-native to the chicken gut (phylum level: cyanobacteria, planctomycetes, dependentiae, verrucomicrobia; order level: rickettsiales, rhodobacterales, holosporales, sphingomonadales, mollicutes RF39, mollicutes incertae sedis) were excluded (2486 ASVs remaining). Moreover, ASVs present in less than two samples (1083 ASVs remaining) and with a total abundance below 0.001% across all samples (910 ASVs remaining) were removed. Rarefaction curves were generated using the *vegan* package, and samples were rarefied to 24,000 reads per sample (910 ASVs remaining) using the *Phyloseq* package. If not stated differently, subsequent analyses were conducted for filtered and rarefied data and for each intestinal segment separately.

Alpha diversity was determined by the Shannon diversity index using the R package *microbiome*. A *t*-test was conducted to assess differences between intestinal segments and a Kruskal–Wallis test was conducted to assess differences between dietary treatments, days, and for dietary treatments within each sampling day, separately, followed by pairwise comparisons using the pairwise Wilcoxon rank sum test applying a Benjamini-Hochberg correction implemented in the R package *stats*.

Beta diversity between dietary treatments was examined performing a Principle Coordinate Analysis (PCoA) on Bray-Curtis dissimilarity distances. Bray-Curtis dissimilarity distances were estimated with *Phyloseq*. To determine differences between dietary treatments and days, a permutational multivariate analysis of variance (PERMANOVA) was performed based on Bray-Curtis distances for dietary treatment and days, separately, using the *adonis* function implemented in *vegan,* applying 999 permutations. Homogeneity of group dispersions was verified using the *betadisper* function implemented in *vegan*. The 10 most abundant bacterial families were determined and visualized as heatmaps using the R package *ampvis2*. Statistical tests were considered significant with *p*-value less than 0.05.

Differential ASV abundance was analyzed for filtered but not rarefied ASV counts aggregated at genus level for ileum and ceca separately using *DESeq2*. ASV counts were normalized applying the variance-stabilizing transformation approach implemented in *DESeq2*. The function *DESeq* implemented in the *DESeq2* package was used to identify differentially abundant ASVs. ASVs were considered differentially abundant for Benjamini-Hochberg [42] adjusted *p*-values ≤ 0.05. Results were visualized using the package *ggplot2*.

## 3. Results

### 3.1. Production Performance

The effect of dietary treatments on broiler performance is shown in Table 1. Compared to broilers fed the non-supplemented control diet, the broilers supplemented with butyrate did not show any improvement in weight gain, feed intake, FCR, nor water to feed intake ratio (*p* > 0.05). Dietary addition of salinomycin resulted in significantly higher body weight, body weight gain, feed intake, and water intake on day 23, 34, and, on average, during the entire experimental period (*p* < 0.001, Table 1). On day 23, salinomycin-fed chickens showed improved FCR as compared to the butyrate-fed chickens (*p* = 0.013). On day 34 and, on average, during the entire experimental period, the best FCR was observed in the salinomycin-fed broilers (*p* = 0.006 and *p* = 0.046, respectively). Moreover, salinomycin significantly reduced the ratio of drinking water to feed intake, on average, during the entire experimental period (*p* = 0.025). The average mortality was 4.5%, and there was no significant difference between treatment groups. At necropsy, no macroscopic intestinal lesions were recorded in any of the groups.

### 3.2. Enumeration of Bacteria

In general, the bacterial populations in ileum and ceca were not affected in butyrate-fed animals compared to the animals fed the non-supplemented control diet (Table 2). There were no significant differences between the dietary treatment groups with respect to the number of lactic acid bacteria and coliform bacteria in the ileum (*p* > 0.05), whereas, in cecal digesta, the numbers of lactic acid bacteria and coliform bacteria were reduced in the salinomycin-fed broilers (*p* = 0.006 and *p* = 0.032, respectively). A significant interaction between dietary treatments and age was found with respect to lactic acid bacteria counts in ileum and ceca (*p* = 0.002 and *p* = 0.004, respectively). The numbers of *L. salivarius* were lower in ileum and ceca of broilers receiving salinomycin (*p* = 0.056 and *p* < 0.001, respectively). Salinomycin-fed broilers showed increased numbers of lactose-negative enterobacteria in the ileum (*p* = 0.011). Likewise, a significant interaction was found between dietary treatments and age with respect to lactose negative enterobacteria counts in ileum and ceca (*p* = 0.014 and *p* = 0.006, respectively). No differences were found between dietary treatment groups with respect to enterococci counts in ileum and ceca. However, a significant interaction between dietary treatments and age was found with respect to enterococci counts in cecal content (*p* = 0.012). In comparison with control and butyrate-fed broilers, the dietary supplementation of salinomycin significantly reduced the numbers of *C*. *perfringens* in the ileum (app. 36% decrease) and ceca (app. 43% decrease) (*p* = 0.003 and *p* < 0.001, respectively). Interaction between dietary treatments and age was found in ceca with respect to *C. perfringens* (*p* = 0.056) and *L. salivarius* (*p* = 0.047). *C. perfringens* numbers in cecal digesta of salinomycin-fed broilers were below or close to the detection limit of log 3 CFU/g digesta.

In ileal digesta, lactic acid bacterial counts decreased gradually from day 21 to 29 (*p* < 0.001), whereas lactic acid bacterial counts in cecal digesta decreased from day 21 to 24 (*p* < 0.001). In both ileum and ceca, the numbers of *L. salivarius* were significantly lower at day 29 compared to other days (*p* < 0.001). The numbers of coliform bacteria in the ileum increased gradually with age, with day 29 being significantly higher than day 21 (*p* = 0.020). In cecal digesta, the numbers of coliform bacteria were lowest at day 21 (*p* < 0.001). The numbers of lactose-negative enterobacteria increased from day 21 to 24 in ceca (*p* < 0.001). Enterococci counts in ceca increased from day 21 to 24 and decreased afterwards (*p* = 0.010). In cecal digesta, the numbers of *C. perfringens* decreased with age and were significantly lower at day 29 (*p* < 0.001), whereas age had no influence on the numbers of *C. perfringens* in ileal contents.

### 3.3. Organic Acids

Treatment and age were found to have a significant effect on SCFA concentration in ileal and cecal digesta (Table 3). In general, the major product of microbial fermentation in the ileum was lactic acid, whereas, in ceca, it was acetic acid, followed by butyric acid. Butyrate-fed and salinomycin-fed broilers showed lower concentration of lactic acid (*p* = 0.036) and succinic acid (*p* = 0.043) in ileal digesta compared to the non-supplemented control. In ileal digesta, significant interactions between dietary treatments and age were observed for lactic acid (*p* = 0.038) and succinic acid (*p* = 0.025), as well as formic acid (*p* = 0.025). Succinic acid in ileal digesta was detected in low concentrations and was below detection level on day 29. Similarly, the concentration of formic acid in cecal digesta was below detection level during the course of the experiment.

In cecal digesta, succinic acid concentration was higher in salinomycin-fed broilers than non-supplemented broilers (*p* < 0.001). The concentration of isobutyric acid and isovaleric acid was highest in butyrate-fed broilers (*p* < 0.001). The concentration of valeric acid in ceca was higher in butyrate-fed broilers, as compared to salinomycin-fed broilers (*p* = 0.003). Interactions between dietary treatments and age were found significant with respect to butyric acid (*p* = 0.009) and succinic acid (*p* = 0.025) in ceca. Lactic acid concentrations in cecal content were very low and no effect of age or treatment on lactic acid concentration was found. Further, the concentration of acetic acid in ileal digesta increased gradually with age (*p* < 0.001). Lactic acid concentration in ileum declined significantly on day 29 (approximately 52% decrease). Similarly, a decline in succinic acid concentration was observed on day 29 in ileum (*p* < 0.001). In cecal digesta, the concentration of propionic acid and valeric acid decreased on day 29 (*p* < 0.001). The concentration of isobutyric acid in ceca was higher on day 24 compared to the other days (*p* < 0.001). In cecal digesta, isovaleric acid concentration increased on day 24 (*p* < 0.001), while succinic acid concentration declined on day 24 and remained stable afterwards (*p* < 0.001).

### 3.4. pH

Treatment and age effects on pH of ileal and cecal digesta are shown in Table 4. No effect of the dietary treatment on ileal pH was found. Ileal pH increased gradually by age (*p* < 0.001). Cecal pH in salinomaycin-fed broiler was significantly lower than that in the control group and in the butyrate-fed group (*p* < 0.01). However, no age effect on pH of cecal digesta was found. Interactions between dietary treatments and age were found with respect to pH in ileum and ceca (*p* = 0.009).

### 3.5. 16S rRNA Gene Amplicons

Sequencing of 142 samples yielded a total of 8,039,711 reads, ranging from 583 to 170,024 reads per sample. A total of 139 samples and 7,258,244 reads remained after filtration. A sequencing depth of 24,000 reads was considered appropriate from rarefaction curves (Appendix A), which resulted in 139 samples, 3,288,000 reads in total, and 910 ASVs for further analyses. The 910 ASVs were taxonomically assigned into 6 phyla, 35 families (1 ASV of unknown family, 99.89% assigned), and 109 genera (86.26% assigned).

#### 3.5.1. Dietary Effect on Microbial Alpha Diversity

To determine the effects of dietary treatments and age on the ileal and cecal microbiota of broilers during intestinal inflammation, the within-community (alpha) diversity was assessed and visualized (Figure 1, Figure 2 and Figure 3). As expected, the microbial communities were dependent on the intestinal segment (Figure 1). When comparing intestinal segments (Figure 1A), regardless of treatment and age, the Shannon diversity was significantly higher in ceca compared to ileum (*p*_t-test_ < 0.001). However, significant differences in the diversity of ileal (*p*_Kruskal–Wallis_ = 0.17) and cecal (*p*_Kruskal–Wallis_ = 0.3) microbiota between the 3 dietary treatments could not be detected (Figure 2B). Likewise, looking at the treatment effect within each segment and for each sampling day separately, no differences could be detected, neither in ileum (Figure 2A) nor cecum (Figure 2B). Further analysis of broiler age on diversity revealed no significant difference in the ileal microbial diversity between the 3 sampling days (*p*_Kruskal–Wallis_ = 0.17) (Figure 1C). In contrast, broiler age showed an influence on Shannon diversity in ceca (*p*_Kruskal–Wallis_ = 0.04) (Figure 1C), where the diversity was higher on day 29 compared with day 21 (*p*_Wilcoxon_ = 0.038). In addition, when comparing age effect within each treatment (Figure 3), age influenced the Shannon diversity only in ceca of butyrate-fed chickens (Kruskal−Wallis, *p* = 0.04).

#### 3.5.2. Dietary Effect on Microbial Beta Diversity

Beta diversity was visualized with PCoA ordination plots based on Bray-Curtis distances. Dietary effects on bacterial composition in ileal contents could not be detected, neither at day 21, 24, nor 29 (Figure 4A–C; Appendix A). In contrast, Dietary treatments effects on beta diversity were detected in cecal contents at days 21 (*p* = 0.043), 24 (*p* = 0.001), and 29 (*p* = 0.002), with samples from salinomycin-fed chickens clustering separately along PCo1 (Figure 4E,F; Appendix A). Dietary treatment explained 10% of the total bacterial variation observed at day 21, 17.16% at day 24, and 15.48% at day 29 (Appendix A). Moreover, differences in bacterial community composition related to broiler age were detected both in cecal and ileal contents and irrespective of dietary treatment supplied (ileum: *p*_butyrate_ = 0.003, *p*_salinomycin_ = 0.001, *p*_control_ = 0.001; ceca: *p*_butyrate_ = 0.001, *p*_salinomycin_ = 0.001, *p*_control_ = 0.001; Appendix A).

#### 3.5.3. Dietary Effect on Microbiota in Different Segments

The 10 most abundant bacterial families are presented in Figure 5 and Figure 6, grouped according to dietary treatments and age within each GI segment, separately. Overall, *Lactobacillus*, *Ruminococcaceae*, and *Lachnospiraceae* were the most dominating families in the intestine (Appendix A). The microbial community in the ileum (Figure 5) was dominated by *Lactobacillaceae*, followed by *Peptostreptococcaceae*, *Enterobacteriaceae*, *Clostridiaceae 1*, *Streptococcaceae*, and *Enterococcaceae,* whereas cecal microbial communities (Figure 6) were dominated by *Ruminococcaceae* and *Lachnospiraceae*, followed by *Clostridiales vadinBB60* group, *Lactobacillaceae*, *Enterobacteriaceae*, *Streptococcaceae*, and *Erysipelotrichaceae*.

Of the 109 genera, a total of ten assigned and two unknown genera belonging to eight different families were significantly affected by at least one treatment (Figure 7). Of the assigned genera, seven were found differentially abundant in ceca samples (*Butyricicoccus*, *Enterococcus*, *Escherichia−Shigella*, *Lachnospiraceae NK4A136*, *Oscillibacter*, *Ruminiclostridium 9*, and *Streptococcus*), one was found differentially abundant only in ileum samples (*Lactobacillus*), and one was found differentially abundant in both GI segments (*Romboutsia*). Moreover, *Enterococcus*, *Lachnospiraceae NK4A136*, *Oscillibacter*, and *Streptococcus* were found differentially abundant more than once, but only in the ceca samples. The majority of genera detected differentially abundant were associated with the salinomycin group, as visualized in Figure 7. Moreover, the largest number of differentially abundant genera was detected in ceca at day 24 (six genera from 5 families) and 29 (six genera from 3 families) when comparing salinomycin- and butyrate-fed chickens.

In general, bacterial abundances in ileal contents seem to be more stable in response to dietary treatments than bacterial abundances in cecal contents, and only two incidences of differentially abundant taxer were found in ileum samples (Figure 7A). Furthermore, dietary treatments mainly affected genera belonging to the Bacilli or Clostridia class. Genera belonging to the Bacilli class (*Enterococcus* and *Streptococcus)* were consistently lower in salinomycin-fed chickens in ceca. In ileum samples, in contrast, the only differential abundant genus belonging to the Bacilli class (*Lactobacillus*) was found to be higher in salinomycin-fed compared to control chickens. Genera belonging to the Clostridia class did not show clear patterns at class level. At family level, however, Lachnospiraceae were reduced in salinomycin-fed chickens, whereas Ruminococcaceae were mainly enriched. *Lachnospiraceae NK4A136*, *Romboutsia*, and a taxon of unknown genus showed largest fold changes (log2 fold change > 5) observed in response to dietary treatment. Moreover, the difference in read abundance of *Lachnospiraceae* NK4A136 (log2 fold change on day 21 = −5.94 versus log2 fold change on day 29 = −7.75) in ceca samples became more prominent over time in salinomycin-fed chickens compared to the control group. The same trend was observed for the difference in read abundance of *Enterococcus* (log2 fold change on day 24 = −2.86 versus log2 fold change on day 29 = −3.49) in salinomycin- compared to butyrate-fed chickens (Figure 7B). Furthermore, results indicate that *Enterococcus* abundances in cecal samples were sensitive to both the salinomycin- and the butyrate treatment, but with opposite effect. Compared to the control group, the *Enterococcus* abundance was lower in salinomycin-fed chickens at day 29 (*p*_adj_ < 0.001; log2 fold change = −3.49) but was higher in butyrate-fed chickens at day 21 (*p* _adj_ = 0.018; log2 fold change = 2.61). Moreover, when comparing salinomycin- with butyrate-fed chickens, the *Enterococcus* abundance was lower in salinomycin-fed chickens on day 24 (*p* _adj_ = 0.001; log2 fold change = −2.86) and day 29 (*p* _adj_ < 0.001; log2 fold change = −4.02).

## 4. Discussion

The beneficial effects of supplementary butyrate feeding on poultry are well-documented and characterized by growth performance improvement and potential positive modulation of GI tract microbiota [16,17,21,43,44,45]. Moreover, the favorable influence of salinomycin has been frequently studied as coccidiostat, as well as an antibacterial agent in poultry feed to control enteric diseases [44,46,47]. However, the effect of dietary supplementation of butyrate and salinomycin on the GI bacterial community composition under the experimental dysbacteriosis in broilers is poorly characterized. Different models of experimental dysbacteriosis have been used previously by inclusion of high levels of rye (20–58%) in the diet [10,18,19,20]. Although these models effectively induced gut leakage and inflammation, they do not realistically reflect practical feeding conditions, where rye is not used in broiler nutrition, and where wheat-based feed is supplemented with enzymes degrading soluble non-starch polysaccharides (NSP), thus preventing intestinal viscosity. In the present experiment, the challenge included a combination of a 10 times overdose of an attenuated coccidiosis vaccine and 5% dietary rye. Intestinal samples were taken from days 21 to 29 of age because intestinal health problems, e.g., dysbacteriosis, are likely to occur at this age [8]. Our previous results demonstrated that this experimental model was able to induce intestinal barrier failure, which was alleviated by salinomycin supplementation [21]. Nevertheless, whether the observed effects are a consequence of the coccidia challenge or the dietary inclusion of 5% rye is unclear. In general, the results of this study are in agreement with previous studies, where the population of *Lactobacillaceae* was mentioned as the most abundant family in the ileum (53–78%), and *Ruminococcaceae* (53–57%) and *Lachnospiraceae* (22–23%) as the most abundant families in the ceca [48,49]. As expected, the microbial communities in ileum and ceca differed from each other, and dietary treatments and age had different effects on the microbial community in the two intestinal segments. In addition, as in previous reports [48,49], the highest bacterial diversity was found in the ceca, this was supported by both higher concentrations and higher diversity of organic acids in cecal contents.

In terms of microbial ecology, dietary treatments had no significant influence on bacterial diversity in ileum and ceca, measured by Shannon diversity index. Moreover, beta diversity analysis revealed comparable bacterial communities in the ileum of broilers in all dietary treatment groups. However, significant differences in cecal bacterial communities were observed related to dietary treatments. Literature indicates that supplementing diets of broilers with butyrate influences cecal microbiota composition in a way that is beneficial for the health and growth performance when the microbiota is disturbed or during enteric disease, e.g., *Salmonella* Enteritidis and *Eimeria maxima* infection [1,16,50,51,52]. Inconsistent with earlier studies, present results revealed dietary supplementation with butyrate had no substantial influence either on broiler performance or on ileal and cecal microbial diversity [44]. On the other hand, butyrate markedly reduced lactic acid concentration in ileum and increased isobutyrate acid and isovaleric acid concentrations in ceca compared to control. Given no difference in performance results between butyrate-fed chicken and non-supplemented control chicken, it is difficult to infer any beneficial effect of the current dietary butyrate concentration on the composition of the intestinal microflora under the chosen challenge condition.

*Enterobacteriaceae* have a specific status in the study of gut inflammation and colitis, with literature reporting their role in inducing intestinal inflammation [53]. They are also known as general indicator of a disrupted intestinal microbiota [43,54]. In our study, salinomycin-fed chickens maintained favorable intestinal environments and health by impairing the growth of *Enterobacteriaceae* abundance in ceca. Potent inflammatory pathogen-associated molecular pattern molecules (PAMPs), such as lipopolysaccharide from *Enterobacteriaceae*, are thought to exacerbate NSAID-induced intestinal injury and increase intestinal permeability in celiac disease in human [53]. This result correlates very well with our recent report, in which the intestinal permeability was higher in the non-supplemented control chickens and in the butyrate-fed chickens in comparison with salinomycin-fed chickens [21]. Thus, we speculated that impaired growth of *Enterobacteriaceae* exerts a beneficial effect on the salinomycin-fed chickens.

Contrary to a previous study [46], the relative abundance of the *Enterococcus* genus was suppressed in salinomycin-fed chickens in comparison with the two other groups. The *Enterococcus* genus, one of the most important lactic acid-producing bacteria, is found as a natural inhabitant of the poultry GIT with some species for example *Enterococcus faecium* being used as a probiotic [2,13,55]. In this particular case, several studies described a protective effect of *Enterococcus faecium* against pathogenic bacteria and viruses [56,57,58], pointing out a potential double-edged sword role of Entrococcus genus in poultry gut health. Given a difference in relative abundance of Entrococcus genus between salinomycin-fed chicken and the two other groups, therefore, it could be speculated that beneficial effect of salinomycin on performance is due to alternation of Enterococcus genus abundance. However, further studies are needed to investigate the role of the Entrococcus genus on incidence of dysbacteriosis in chickens.

However, it is worth mentioning that ionophore-resistant enterococci are a growing threat to nosocomial infection in humans, as well as the dissemination of high degree antibiotic resistance genes [59].

The ceca is by far the most densely colonized microbial habitat in chickens, which is mainly occupied by *Clostridium*, *Lactobacillus*, and *Ruminococcus* [48]. The majority of members of the order Clostridiales in the cecum belong to 3 main families: Clostridiaceae, Lachnospiraceae, and Ruminococcaceae [60]. Of these, the Ruminococcaceae and Lachnospiraceae members produce butyric, as well as formic, acids [1,48], which play a crucial role in the limitation of pathogenic bacteria proliferation and are necessary for the health of epithelial tissue [61]. The Lachnospiraceae community not only degrades fibrous material but also utilizes starch and non-starch polysaccharides [62,63]. Similar to a recent study [64], salinomycin was found to reduce the abundance of the Lachnospiraceae family in cecal content on day 21 and 29. As in previous reports [64], the abundance of the Ruminococcaceae community in the ceca of broiler increased after coccidiostat addition. However, inconsistent to our study, Trela et al. [65] reported a decrease in the abundance of the Ruminococcaceae members and an increase in the abundance of Lachnospiraceae members in the ceca of broilers after salinomycin supplementation. Overall, given the improved performance in salinomycin-fed chickens, an enhanced abundance of Ruminococcaceae cannot be concluded as a negative effect of salinomycin in this case.

Salinomycin inhibited the growth of *C. perfringens* in ileum and ceca, in accordance with earlier studies [41,42]. It is well-known that improving bird performance via supplementation of the ionophore coccidiostat salinomycin is closely related to modulation of the GI microbiota, particularly gram positive bacteria [26,27]. *C. perfringens* is the causative agent of necrotic enteritis and is considered to play an important role in the development of dysbacteriosis and growth depression caused by production of potent toxins [66]. *C. perfringens* numbers were generally rather low in the current experiment as compared to previous reports [11,22,23]. Intestinal infection with coccidia has been identified as an important predisposing factor for *C. perfringens* proliferation [67] and necrotic enteritis. Considering the challenge in form of an attenuated live vaccine used in the present disease model, the low counts of *C. perfringens* in the current study are somewhat surprising. Furthermore, *L. salivarius*, a dominant lactic acid bacterium in the chicken intestine, was inhibited in ileal and cecal contents following addition of salinomycin [41]. Lactobacilli are known to be producers of bile salt hydrolase responsible for hydrolyzing and de-conjugating primary bile acids [53]. Similarly, *C. perfringens* have been shown to express high levels of bile salt hydrolase activity in the small intestine of broiler chickens [68]. These effects may collectively contribute to the improved performance of broilers supplemented with salinomycin.

SCFAs are produced by bacterial fermentation and play a critical role in mucosal integrity, local and systemic metabolic function, and regulation of immune response. Compared to the control group, there was no significant difference in ileal lactic acid concentration in salinomycin-fed chickens and butyrate-fed chickens. This likely reflects the lactic acid bacterial counts in the ileum, which followed the same pattern. Inconsistent with our study, reduced lactic acid concentration in the ileum following salinomycin supplementation has previously been reported, most likely due to an inhibition of certain Lactobacillus species [1,13,23]. As in previous reports [49], a major decline in ileal lactic acid concentrations occurred on day 29 most likely related to the numbers of lactic acid bacteria and *L. salivarius* in ileal contents, which were lowest on day 29. Isobutyric acid and isovaleric acid derive from the fermentation of amino acids valine and leucine, respectively, originating from undigested protein reaching the ceca [1,69]. Thus, the lower cecal concentrations of isobutyric and isovaleric acid in broilers fed salinomycin indicate less cecal protein fermentation [69], possibly due to a higher small intestinal protein absorption resulting in lower amounts of undigested dietary protein reaching the ceca. The lower concentration of succinic acid in ceca of salinomycin-fed broilers, compared to the non-supplemented control, may be due to the reduced abundance of the *Lachnospiraceae* community in ceca, which are responsible for degradation of fibrous material; therefore, more fiber is available for succinate producing microbiota.

In agreement with our result, the observed pH values reflect SCFA concentrations. Particularly in ceca, marked increase in total SCFA were observed with concomitant effects on digesta pH where salinomycin-fed chickens reduced pH, thus establishing a friendly environment for the proliferation of *Ruminococcaceae* and *Lactobacillaceae* populations as the dominant community in ceca [1]. The opposite effect was noted regarding ileal pH, where, despite higher lactic acid concentration in butyrate-fed chickens, no reduction of pH was found.

These results collectively suggest that improved growth and feed utilization imposed by salinomycin in broilers challenged with an overdose of an attenuated live coccidiosisvaccine and dietary inclusion of rye (5%) could be explained by (1) impaired growth of *Enterobacteriaceae* which play a role in disrupting intestinal digestion and absorption, (2) the suppression of gram-positive bacteria, such as *C. perfringens*, involved in necrotic enteritis and dysbacteriosis [44], and (3) the tendency in enriching butyrate- and lactic acid-producing bacteria (*Ruminococcaceae* and *Lactobacillaceae*), while suppressing bile salt hydrolase-producing bacteria, particularly *L. salivarius* [47,70] and *C. perfringens* [71], although they are yet to be experimentally verified.

Taken together, our results support the importance of salinomycin in maintaining the intestinal structure and biological functions of broilers through modulation of the microbial community under enteric challenge condition. These results are, to our knowledge, the first in vivo data documenting an ecological effect of salinomycin on microbial community in enteric challenge condition, with important consequences for gut health.

## 5. Conclusions

It was shown that salinomycin could alleviate growth depression caused by experimental enteric challenge and maintain physiological homeostasis in broilers by modulation of intestinal microbial composition, while the dietary inclusion of coated butyrate (0.5 g/kg feed) showed no or limited effects. Thus, further investigations are required to identify optimal dietary inclusion rates for coated butyrate used as alternative to ionophore coccidiostats in broiler production.

## Figures and Tables

**Figure 1 animals-12-00111-f001:**
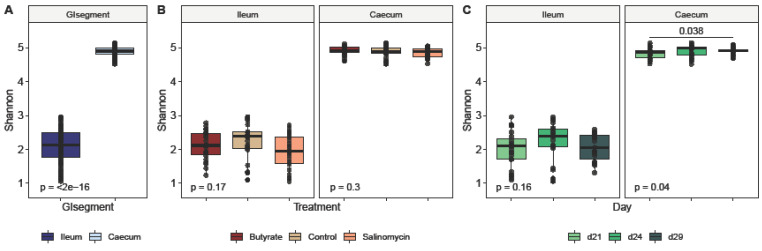
Alpha diversity based on Shannon Index. (**A**) GI segment effect on alpha diversity (all samples combined), (**B**) treatment effect on alpha diversity (for all ceca samples combined (left graph, see title) and all ileum samples combined (right graph, see title), and (**C**) age effect on alpha diversity (for all ceca samples combined (left graph, see title) and all ileum samples combined (right graph, see title)). P-values provided: For (**A**) differences between GI segments were identified by *t*-test. For (**B**,**C**), overall treatment and age effect, respectively, were identified using a Kruskal–Wallis test (provided in the bottom of each graph), and pairwise differences between treatment or age groups, respectively, were determined by pairwise Wilcoxon rank sum tests (provided as horizontal bars). Only significant pairwise differences are visualized.

**Figure 2 animals-12-00111-f002:**
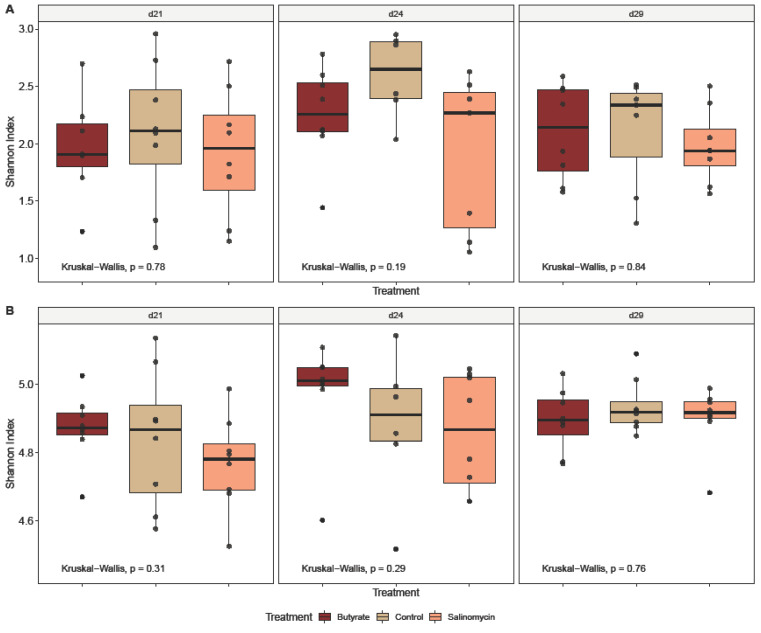
Treatment effect on alpha diversity (Shannon Index) for (**A**) ileum and (**B**) ceca samples collected at days 21, 24, and 29 of age from chickens administered butyrate (UltraGuard™-DUO, 500 mg/kg), salinomycin (Sacox^®^, 69 mg/kg), or no dietary supplement (control). Overall treatment effects were identified using a Kruskal–Wallis test (provided in the bottom of each graph) and pairwise differences between treatments were determined by pairwise Wilcoxon rank sum tests (provided as horizontal bars with *p* values). Only significant pairwise differences are visualized.

**Figure 3 animals-12-00111-f003:**
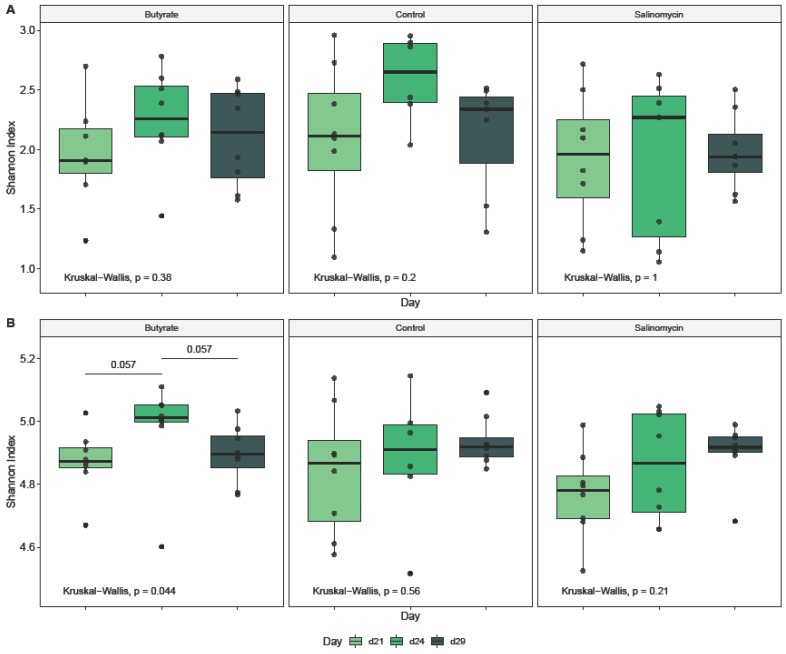
Broiler age effect on alpha diversity (Shannon Index) for (**A**) ileum and (**B**) ceca samples collected from chickens administered butyrate (UltraGuard™-DUO, 500 mg/kg), salinomycin (Sacox^®^, 69 mg/kg), or no dietary supplement (control) and investigated for each treatment group separately. Overall age effects were identified using a Kruskal–Wallis test (provided in the bottom of each graph), and pairwise differences between age groups were determined by pairwise Wilcoxon rank sum tests (provided as horizontal bars with *p* value). Only significant pairwise differences are visualized.

**Figure 4 animals-12-00111-f004:**
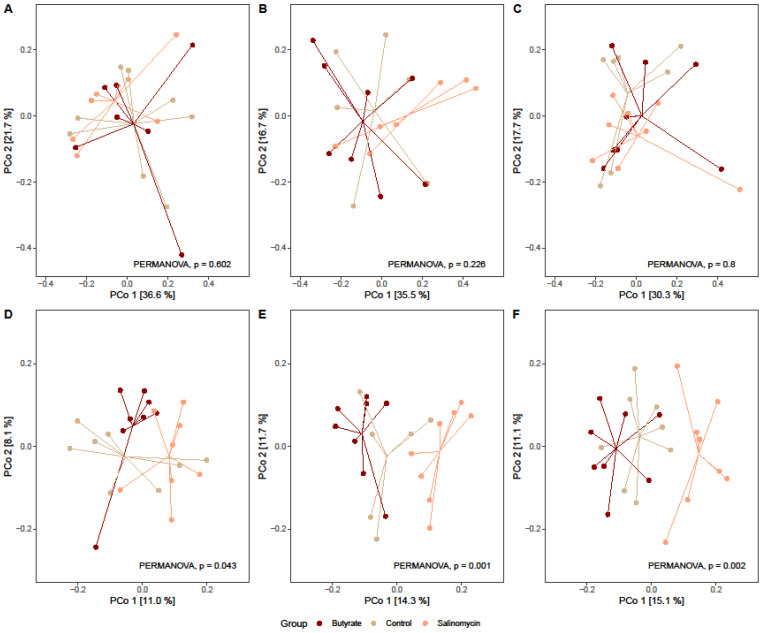
Treatment effect on beta diversity (Bray-Curtis distances) for ileum samples collected on days (**A**) 21, (**B**) 24, and (**C**) 29, and for ceca samples collected on days (**D**) 21, (**E**), 24, and (**F**) 29 of age from chickens administered butyrate (UltraGuard™-DUO, 500 mg/kg), salinomycin (Sacox^®^, 69 mg/kg), or no dietary supplement (control). Overall differences in centroids (group means) were estimated by PERMANOVA, and *p* values are provided.

**Figure 5 animals-12-00111-f005:**
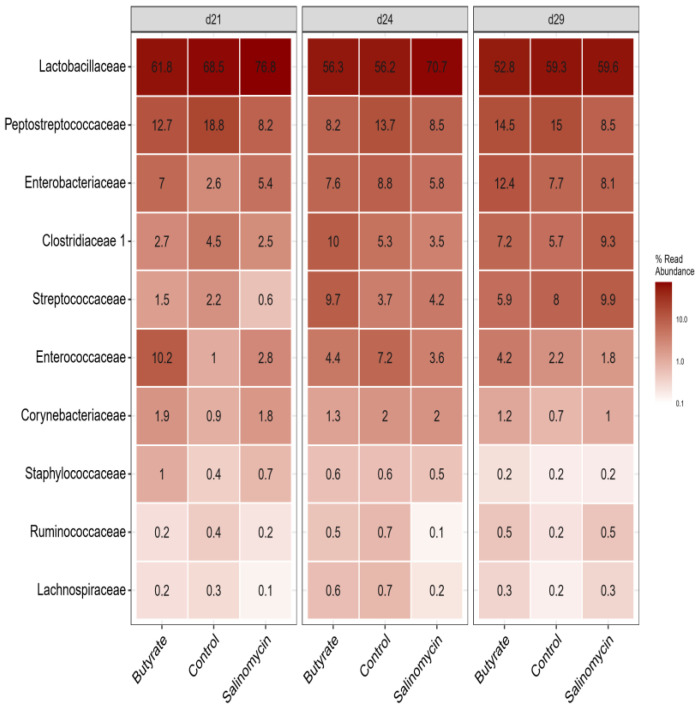
Heatmap showing the relative abundance of the top 10 most abundant microbiota at the family level across all ileum samples (irrespective of treatment or age) and visualized for each age and treatment group, separately. The samples were collected on days 21, 24, and 29 of age from chickens administered butyrate (UltraGuard™-DUO, 500 mg/kg), salinomycin (Sacox^®^, 69 mg/kg), or no dietary supplement (control).

**Figure 6 animals-12-00111-f006:**
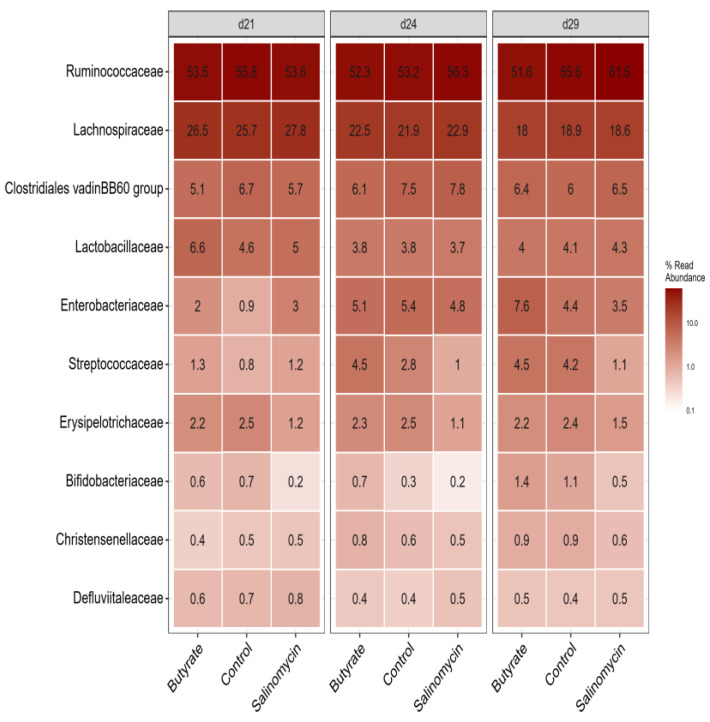
Heatmap showing the relative abundance of the top 10 most abundant microbiota at the family level across all ceca samples (irrespective of treatment or age) and visualized for each age and treatment group, separately. The samples were collected on days 21, 24, and 29 of age from chickens administered butyrate (UltraGuard™-DUO, 500 mg/kg), salinomycin (Sacox^®^, 69 mg/kg), or no dietary supplement (control).

**Figure 7 animals-12-00111-f007:**
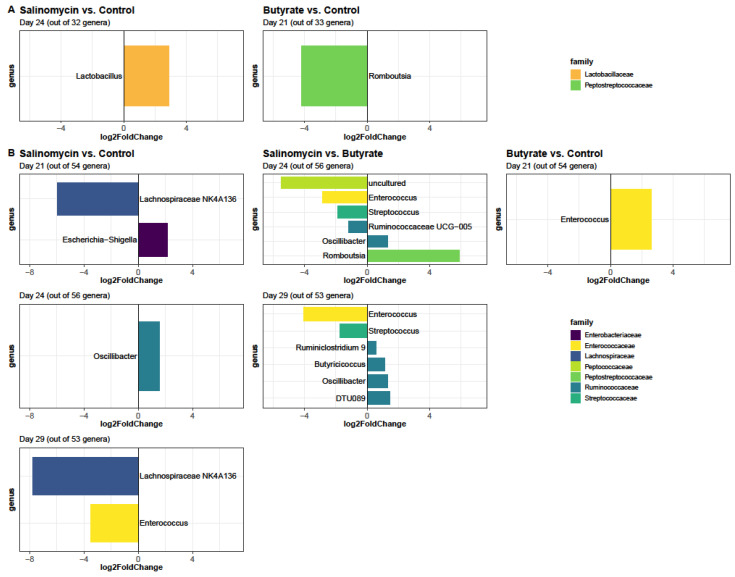
Differential abundance analysis at genus level for (**A**) ileum and (**B**) ceca samples between butyrate vs. control, salinomycin vs. control, and butyrate versus salinomycin (referred to as contrasts). Contrasts were analyzed for each sampling day separately. A genus was define differentially expressed for Benjamini-Hochberg adjusted *p* values ≤ 0.05. Only significant results are shown. Information on contrasts, sampling day and total number of detected ASVs are defined in each figure title. A positive fold change means that the taxa abundance was higher in the first named treatment group compared to the second named treatment group (e.g., for the comparison butyrate versus control, taxa with positive fold changes are higher expressed in butyrate compared to control samples collected at the same day). A negative fold change means that the taxa abundance was lower in the first named treatment group compared to the second named treatment group. Color scale applied represents the family assignment of each differentially abundant ASVs.

**Table 1 animals-12-00111-t001:** Effect of dietary treatments on body weight, body weight gain, feed intake, water intake, feed conversion ratio (FCR), and water/feed intake ratio in broilers at different ages ^1^.

	Dietary Treatment					
Item	Age (day)	Control	Butyrate	Salinomycin	SEM	*p* Value
**Body weight (g/day/bird)**	10	249	253	260	3.1	>0.05
	23	653 ^b^	673 ^b^	770 ^a^	16.6	<0.001
	34	787 ^b^	822 ^b^	1104 ^a^	25.0	<0.001
**Body weight gain (g/day)**	1–10	24.9	25.3	26	0.31	0.063
	11–23	50.2 ^b^	51.8 ^b^	59.2 ^a^	1.27	<0.001
	24–34	71.6 ^b^	74.7 ^b^	100.4 ^a^	2.27	<0.001
	1–34	59.7 ^b^	61.9 ^b^	72.8 ^a^	1.66	<0.001
**Feed intake (g/day/bird)**	1–10	29.7	28.9	29.1	0.88	>0.05
	11–23	73.6 ^b^	78.5 ^b^	83.8 ^a^	1.63	<0.001
	24–34	119 ^b^	123 ^b^	148 ^a^	3.77	<0.001
	1–34	78.7 ^b^	81.8 ^b^	92.2 ^a^	1.78	<0.001
**Water intake (mL/day/bird)**	1–10	61.8	62.2	62.5	1.18	>0.05
	11–23	140 ^b^	146 ^ab^	154 ^a^	3.26	0.024
	24–34	198 ^b^	207 ^b^	238 ^a^	6.57	<0.001
	1–34	134 ^b^	139 ^b^	152 ^a^	3.12	0.002
**FCR** **(kg diet/kg weight gain)**	1–10	1.19	1.14	1.10	0.026	0.076
	11–23	1.47 ^ab^	1.52 ^a^	1.42 ^b^	0.021	0.013
	24–34	1.66 ^a^	1.64 ^a^	1.47 ^b^	0.044	0.006
	1–34	1.32 ^a^	1.32 ^a^	1.27 ^b^	0.016	0.046
**Water/feed intake ratio**	1–10	2.11	2.15	2.15	0.065	>0.05
	11–23	1.91 ^ab^	1.87 ^b^	2.01 ^a^	0.035	0.022
	24–34	1.67 ^b^	1.70 ^b^	1.93 ^a^	0.055	0.005
	1–34	1.71 ^a^	1.71 ^a^	1.65 ^b^	0.016	0.025

^1^ Values are presented as least square means and standard error of mean (SEM) with a total of 8 replicate pens in each treatment group. Pen was considered as an experimental unit. ^a,b^ Means within the same row with different superscript letters differ significantly (*p* ≤ 0.05).

**Table 2 animals-12-00111-t002:** Bacterial counts (log CFU/g digesta) in the ileum and ceca of broilers at different ages, fed a diet containing salinomycin, coated butyrate, or no dietary supplement (control) ^1^.

	Dietary Treatment				Age (day)				*p* Value		
Item	Control	Butyrate	Salinomycin	SEM	21	24	29	SEM	Diet	Age	Diet × Age
** *Lactic acid bacteria* **											
Ileum	8.47	8.23	8.36	0.09	8.62 ^a^	8.25 ^bc^	8.05 ^c^	0.09	0.181	<0.001	0.004
Ceca	9.11 ^a^	9.14 ^a^	8.93 ^b^	0.05	9.30 ^a^	9.02 ^b^	8.89 ^b^	0.06	0.006	<0.001	0.002
** *Coliform bacteria* **											
Ileum	6.37	6.48	6.63	0.14	6.19 ^b^	6.54 ^ab^	6.48 ^ab^	0.14	0.362	0.020	0.102
Ceca	8.61 ^a^	8.69 ^a^	8.44 ^b^	0.07	8.07 ^b^	8.84 ^a^	8.76 ^a^	0.08	0.032	<0.001	0.342
** *Lactose-negative enterobacteria* **											
Ileum	4.66 ^b^	4.46 ^b^	5.03 ^a^	0.14	4.27 ^b^	4.29 ^b^	4.72 ^b^	0.14	0.011	<0.001	0.014
Ceca	6.45	6.46	6.44	0.14	6.00 ^b^	6.72 ^a^	6.54 ^a^	0.11	1.000	<0.001	0.006
** *Enterococci* **											
Ileum	6.75	6.77	6.97	0.10	6.70 ^b^	6.79 ^ab^	6.72 ^b^	0.10	0.222	0.009	0.332
Ceca	7.46	7.45	7.30	0.08	7.31 ^b^	7.68 ^a^	7.34 ^ab^	0.09	0.289	0.010	0.012
** *Clostridium perfringens* **											
Ileum	3.20 ^a^	3.39 ^a^	2.05 ^b^	0.28	2.79	3.13	2.78	0.25	0.003	0.621	0.351
Ceca	4.11 ^a^	4.41 ^a^	2.34 ^b^	0.35	4.31 ^a^	3.87 ^a^	3.49 ^ab^	0.28	<0.001	<0.001	0.056
** *Lactobacillus salivarius* **											
Ileum	7.40	7.27	6.95	0.14	7.33 ^a^	7.49 ^a^	6.58 ^b^	0.13	0.056	<0.001	0.851
Ceca	8.39 ^a^	8.45 ^a^	8.01 ^b^	0.07	8.58 ^a^	8.48 ^a^	7.62 ^b^	0.08	<0.001	<0.001	0.047

^1^ Values are presented as least square means and standard error of mean (SEM) with a total of 8 replicate pens in each treatment group. Pen was considered as an experimental unit. ^a,b,c^ Means within the same row with different superscript letters differ significantly (*p* ≤ 0.05).

**Table 3 animals-12-00111-t003:** Concentration (mmol/kg digesta) of organic acids in the ileum and ceca of broilers at different ages, fed a diet containing salinomycin, coated butyrate, or no dietary supplement (control) ^1^.

	Dietary Treatment				Age (day)				*p*-Value		
Item	Control	Butyrate	Salinomycin	SEM	21	24	29	SEM	Diet	Age	Diet × Age
** *Ileum* **											
Lactic acid	40.9 ^a^	28.9 ^b^	31.6 ^ab^	5.24	26.9 ^bc^	37.4 ^b^	18.0 ^c^	5.61	0.036	<0.001	0.038
Acetic acid	4.8	4.4	4.78	0.29	3.30 ^c^	4.47 ^b^	4.70 ^b^	0.33	0.13	<0.001	0.12
Formic acid	0.58	0.58	0.57	0.08	0.18 ^c^	0.50 ^bc^	0.54 ^b^	0.09	0.99	<0.001	0.025
Succinic acid	0.31 ^a^	0.22 ^ab^	0.15 ^b^	0.06	0.28 ^a^	0.31 ^a^	0.00 ^b^	0.06	0.043	<0.001	0.025
** *Ceca* **											
Acetic acid	75.3	77.1	81.9	2.22	77.2 ^b^	71.9 ^b^	74.1 ^b^	2.56	0.082	<0.001	0.19
Succinic acid	5.29 ^b^	5.59 ^ab^	8.57 ^a^	0.87	10.3 ^a^	5.98 ^b^	5.76 ^b^	0.97	<0.001	<0.001	0.025
Propionic acid	4.03	4.59	4.34	0.17	4.21 ^bc^	4.28 ^b^	3.51 ^c^	0.2	0.06	<0.001	0.17
Isobutyric acid	0.54 ^b^	0.69 ^a^	0.44 ^c^	0.03	0.45 ^b^	0.66 ^a^	0.48 ^b^	0.03	<0.001	<0.001	0.089
Butyric acid	18.5	18.6	21.1	0.9	19.5 ^b^	17.6 ^b^	16.7 ^b^	1.03	0.054	<0.001	0.009
Isovaleric acid	0.26 ^b^	0.35 ^a^	0.20 ^b^	0.02	0.23 ^b^	0.34 ^a^	0.23 ^b^	0.02	<0.001	<0.001	0.45
Valeric acid	1.02 ^ab^	1.16 ^a^	0.96 ^b^	0.04	1.02 ^ab^	1.08 ^a^	0.90 ^b^	0.04	0.003	<0.001	0.07
Lactic acid	0.66	0.28	0.46	0.29	0.17	1.02	0.37	0.33	0.61	0.25	0.43

^1^ Values are presented as least square means and standard error of mean (SEM) with a total of 8 replicate pens in each treatment group. Pen was considered as an experimental unit. ^a,b,c^ Means within the same row with different superscript letters differ significantly (*p* ≤ 0.05).

**Table 4 animals-12-00111-t004:** pH of digesta from the ilea and ceca of broilers at different ages ^1^.

Item	Dietary Treatment			SEM	Age (day)			SEM	*p*-Value		
	Control	Butyrate	Salinomycin		21	24	29		Diet	Days	Diet x Days
** *Ileum* **	6.30	6.45	6.54	0.10	6.18	6.34	6.69	0.09	0.20	<0.001	0.02
* **Ceca** *	6.43 ^a^	6.38 ^a^	6.09 ^b^	0.06	6.10	6.42	6.34	0.07	<0.001	0.780	0.001

^1^ Values are presented as least square means and standard error of mean (SEM) with a total of 8 replicate pens in each treatment group. Pen was considered as an experimental unit. ^a,b^: Rows with different letters are significantly different (*p* < 0.05).

## Data Availability

Data are available upon request.

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
