# Peer review of "Impact of Dietary Sodium Butyrate and Salinomycin on Performance and Intestinal Microbiota in a Broiler Gut Leakage Model"

_animals, 2022, doi:10.3390/ani12010111_

Round 1

Reviewer 1 Report

The study presented in the manuscript investigated dietary supplementation of salinomycin (a common ionophore coccidiostat) and butyrate (a potential alternative to antibiotics) on intestinal microbiota in a chicken gut leakage model. The topic meets the Aims and Scope of Animals and attracts a wide readership. The Abstract is sufficiently informative and complete. The materials and methods described allow reproducibility, while the majority of results were interpreted appropriately and provide new scientific information on mechanisms of salinomycin to the research field.

Specific comments

  1. In the Simple Summary section, major results on gut microbiota in the treatments of salinomycin and butyrate should be mentioned (Lines 22-24) before stating the conclusion on the link between improved growth and altered gut microbiota for salinomycin.
  2. Dysbacteriosis is a form of alternation in gut microbiota, for example, gut microbiota imbalance. Dysbacteriosis does not equal gut leakage, which, however, could be related to gut leakage. Gut leakage could be a result or cause of dysbacteriosis. So please delete ‘gut leakage’ in Line 30. Meanwhile, please be prudent with the use of dysbacteriosis in other parts of this manuscript, for example, in Line 102, Line 134, Line 512, Line 652.
  3. In the abstract, the overall results of gut microbiota and organic acids in the treatment of butyrate should be mentioned in the abstract although the effects are minimal.
  4. Please check whether the collection of microorganisms is termed ‘microbiome’ or ‘microbiota’ (Line 64).
  5. Gut inflammation and barrier dysfunction are possible consequences of dysbacteriosis, so please consider rewriting the sentence ‘An imbalance in the gut microbial community known as dysbacteriosis includes gut inflammation and barrier dysfunction…’ in Line 73.
  6. Please check the grammar of the sentence ‘Salinomycin is an ionophore coccidiostat, …, among others the intestinal pathogens…’ (from Line 82 to Line 85).
  7. In the sentence of ‘In addition to the role of butyrate in…, however, contrasting results exist.’, change ‘butyrate anti-inflammatory properties in chicken have been’ (Line 96) to ‘anti-inflammatory properties of butyrate in chickens…’. ‘however, contrasting results exist’ can be deleted, otherwise, references for this statement are needed.
  8. Check the grammar of ‘This is while, Chen et al….’ in Line 107.
  9. Was Salinomycin or butyrate given to chickens from day 0? Please clarify in 2.1. Experimental design.
  10. Were body weight, water consumption, and feed intake recorded daily or weekly for growth performance evaluation? Please be specific in Methods (Lines 137 to 139).
  11. The method and equipment to measure the pH of intestinal samples should be specified (Line 149).
  12. Brand information on different agars for bacterial culture should be provided in section 2.3. Bacterial enumeration.
  13. What molecular water (Molecular Grade Water?) was used for DNA dilution (Lines 195 and 196)? Please clarify.
  14. ‘16. S rRNA gene amplicon sequencing’ should be ‘16S rRNA gene amplicon sequencing’ in Line 198?
  15. Should ‘bacterial plate counts’ be ‘bacterial counts’ in Line 218?
  16. It is better to change the word ‘growth performance’ to ‘weight gain’ in Line 291 and in the title of Table 1 because growth performance is a general concept that can be measured by several specific parameters such as weight gain, feed intake, and FCR.
  17. The number of replicates for each treatment should be included in the notation of tables.
  18. The phenotype of this gut leakage model the authors developed should be mentioned in the Results.
  19. In Figure 1, what is the rationale to perform a parametric test (i.e., t-test) to compare the Shannon index between the two segments while using the non-parametric test (i.e., Kruskal–Wallis test) for treatments and age?
  20. In Figures 5 and 6, it would be better to make the font size of treatment and bacterial names larger to facilitate visualization.

Author Response

Reviewer 1

  1. In the Simple Summary section, major results on gut microbiota in the treatments of salinomycin and butyrate should be mentioned (Lines 22-24) before stating the conclusion on the link between improved growth and altered gut microbiota for salinomycin.

A: Both abstract and summary now revised. Lines 16-51

  1. Dysbacteriosis is a form of alternation in gut microbiota, for example, gut microbiota imbalance. Dysbacteriosis does not equal gut leakage, which, however, could be related to gut leakage. Gut leakage could be a result or cause of dysbacteriosis. So please delete ‘gut leakage’ in Line 30.

A: “Gut leakage” is now deleted

  1. Meanwhile, please be prudent with the use of dysbacteriosis in other parts of this manuscript, for example, in Line 102, Line 134, Line 512, Line 652.

A: We have now changed the model conditions to “enteric challenge” and avoided the term “dysbacteriosis” throughout the manuscript.

  1. In the abstract, the overall results of gut microbiota and organic acids in the treatment of butyrate should be mentioned in the abstract although the effects are minimal.

A: The minimal effect of butyrate is now mentioned in the a results section.

  1. Please check whether the collection of microorganisms is termed ‘microbiome’ or ‘microbiota’ (Line 64).

A: Now corrected to “microbiota” in lines 70-71

  1. Gut inflammation and barrier dysfunction are possible consequences of dysbacteriosis, so please consider rewriting the sentence ‘An imbalance in the gut microbial community known as dysbacteriosis includes gut inflammation and barrier dysfunction…’ in Line 73.

A: The sentence is now corrected from line 72-75.

  1. Please check the grammar of the sentence ‘Salinomycin is an ionophore coccidiostat, …, among others the intestinal pathogens…’ (from Line 82 to Line 85).

A: The sentence is now corrected line 65 to 69.

  1. In the sentence of ‘In addition to the role of butyrate in…, however, contrasting results exist.’, change ‘butyrate anti-inflammatory properties in chicken have been’ (Line 96) to ‘anti-inflammatory properties of butyrate in chickens…’. ‘however, contrasting results exist’ can be deleted, otherwise, references for this statement are needed.

A: The sentence is now corrected from line

  1. Check the grammar of ‘This is while, Chen et al….’ in Line 107.

A: The sentence is now corrected from line 100-102

  1. Was Salinomycin or butyrate given to chickens from day 0? Please clarify in 2.1.

A: Day 1 and now clarified in the text.

  1. Experimental design. Were body weight, water consumption, and feed intake recorded daily or weekly for growth performance evaluation? Please be specific in Methods (Lines 137 to 139).

A: Weekly is now specified in the text from line 139 to141.

  1. The method and equipment to measure the pH of intestinal samples should be specified (Line 149).

A: pH meter brand now mentioned from 149 to 151.

  1. Brand information on different agars for bacterial culture should be provided in section 2.3. Bacterial enumeration.

A: Now specified from 161 to 172.

  1. What molecular water (Molecular Grade Water?) was used for DNA dilution (Lines 195 and 196)? Please clarify.

A: Product name and manufacturer now stated in text line 197

  1. ‘16. S rRNA gene amplicon sequencing’ should be ‘16S rRNA gene amplicon sequencing’ in Line 198?

A: Now corrected

  1. Should ‘bacterial plate counts’ be ‘bacterial counts’ in Line 218?

A: Yes, now corrected

  1. It is better to change the word ‘growth performance’ to ‘weight gain’ in Line 291 and in the title of Table 1 because growth performance is a general concept that can be measured by several specific parameters such as weight gain, feed intake, and FCR.

A: Yes indeed, now corrected in line 292.

  1. The number of replicates for each treatment should be included in the notation of tables.

A: Indeed – now corrected

  1. The phenotype of this gut leakage models the authors developed should be mentioned in the Results.

A: No lessions were detected – now clarified in results section from line 302 to 306.

  1. In Figure 1, what is the rationale to perform a parametric test (i.e., t-test) to compare the Shannon index between the two segments while using the non-parametric test (i.e., Kruskal–Wallis test) for treatments and age?

A: Parametric tests (e.g., t-test) are normally more powerful but assume a normal distribution. Normality of alpha diversities was tested for the different data subsets and normality for Shannon Diversity index was violated in the data-subsets used to investigate treatment and age effects therefore we used non-parametric tests instead (Kruskal–Wallis test). We can use non-parametric tests for segments if wished. Results should not differ that much.

  1. In Figures 5 and 6, it would be better to make the font size of treatment and bacterial names larger to facilitate visualization.

A: Figures 5 and 6 font size now are bigger and easy to read

Reviewer 2 Report

Overview: in this manuscript, the authors provided useful information regarding the potential use of coated butyrate and salinomycin as feed additives in broiler diets. The manuscript is well written and well presented. However, some concerns need to be addressed as follows:

  • The title needs to be representative of the experimental design and the aim of the study.
  • According to the journal instructions, the abstract and the simple summary should consist of no more than two hundred words.
  • The objective and the hypothesis of the study should be clearly stated at the end of the Introduction section.
  • Throughout the manuscript, Abbreviations should be defined at first mention and used consistently after that. For example, the full term of FITC_D (Line 113) and IFN-α (Line 114) must be introduced, followed by its abbreviation in parentheses. From then on, the abbreviation must be used exclusively and throughout.
  • Justification of tested additives doses is highly recommended to be clarified with the appropriate references.
  • Line 138: how was the drinking water of the chickens measured?
  • Line 143: Why were close time intervals (day 21 to 29 of age) chosen? Are these time periods representative of the whole experiment?
  • Throughout the results section, I recommend removing all sentences that are no significant effect was observed. The interaction effect was mentioned as significantly affected without details, and this may be because the interaction data (the actually collected data) is not provided. So, it is highly recommended to present these data to give a complete picture of the effect of the tested additives.
  • Is there is any mortality throughout the experiment?
  • In Table 1, and throughout the manuscript, for the beginning of the experiment, instead of zero-0, use one-1, e.g., days 1-10, 11-23, 24-34, and 1-34. Also, The letters of significance for "Water/feed intake ratio" need to be revised on days 1-34.
  • In table 1 also add the data of body weight and water consumption during the experiment intervals.
  • Lines 295-296: In FCR, there were no significant differences between the tested additives and the control group. The significant difference was between the two additives. Please revise and correct.
  • In all Tables footnotes, clarify the experimental unit used and the number of replicates per treatment.
  • The conclusion should be rewritten, as in the current form, it is only a general statement.

Author Response

Reviewer 2

Overview: in this manuscript, the authors provided useful information regarding the potential use of coated butyrate and salinomycin as feed additives in broiler diets. The manuscript is well written and well presented. However, some concerns need to be addressed as follows:

  1. The title needs to be representative of the experimental design and the aim of the study.

A: The title is now corrected

  1. According to the journal instructions, the abstract and the simple summary should consist of no more than two hundred words.

A: abstract and simple summary are now revised

  1. The objective and the hypothesis of the study should be clearly stated at the end of the Introduction section.

A: Now stated from line 110 to 117

  1. Throughout the manuscript, Abbreviations should be defined at first mention and used consistently after that. For example, the full term of FITC_D (Line 113) and IFN-α (Line 114) must be introduced, followed by its abbreviation in parentheses. From then on, the abbreviation must be used exclusively and throughout.

A: Now corrected

  1. Justification of tested additives doses is highly recommended to be clarified with the appropriate references.

A: Both the Salinomycin and butyrate products were used in doses recommended by the companies which is now stated in the text line 127 to 129.

  1. Line 138: how was the drinking water of the chickens measured?

A: This is now specified in the text from line 139-141.

  1. Line 143: Why were close time intervals (day 21 to 29 of age) chosen? Are these time periods representative of the whole experiment?

A: The intestinal imbalances occur in production settings around week 3 of life as referenced in the introduction. We have used a model with Eimeria challenge (on day 17) previously in connection with a necrotic enteritis model and observed intestinal changes and lesions in the interval from day 21-29  – the previous study is now referenced in the introduction (67) in line 105.

  1. Throughout the results section, I recommend removing all sentences that are no significant effect was observed. The interaction effect was mentioned as significantly affected without details, and this may be because the interaction data (the actually collected data) is not provided. So, it is highly recommended to present these data to give a complete picture of the effect of the tested additives.

A: We prefer maintaining information about the not significant data. As for the interactions, we agree that the affected interaction data, would be interesting to know in detail. Please consider providing the interaction data between dietary treatment and age in different segments on bacterial counts, concentrations of organic acids and PH will require several tables. Moreover, these types of interactions cannot easily be inferred by readers and will not help readers to interpret tables 2, 3 and 4 . Importantly, our focus is effect of feed additives on performance of broiler chicken under gut challenge condition where no or limited effects of the dietary inclusion of coated butyrate on broilers physiological homeostasis and intestinal microbial composition were observed. In the light of above-mentioned reasons, we would like to keep the interaction data simple, rather than all in details.

  1. Is there is any mortality throughout the experiment?

A: Mortality did not differ between groups (approx 10/224 in each died during the experiment) this is now stated in the results section. Lines 303 to 306

  1. In Table 1, and throughout the manuscript, for the beginning of the experiment, instead of zero-0, use one-1, e.g., days 1-10, 11-23, 24-34, and 1-34. Also, The letters of significance for "Water/feed intake ratio" need to be revised on days 1-34.

A:The table is now corrected.

  1. In table 1 also add the data of body weight and water consumption during the experiment intervals.

A: The suggested data are now added to table 1.

  1. Lines 295-296: In FCR, there were no significant differences between the tested additives and the control group. The significant difference was between the two additives. Please revise and correct.

A: the reviewer is correct, on day 23 it is only between treatment groups that differences are seen which we have tried to detail like this:  On day 23, salinomycin-fed chickens showed improved FCR as compared to the butyrate-fed chickens (P = 0.013). On day 34 and on average during the entire experimental period, the best FCR was observed in the salinomycin-fed broilers (P = 0.006 and P = 0.046, respectively). line 298 to 302.

  1. In all Tables footnotes, clarify the experimental unit used and the number of replicates per treatment.

A: This is now corrected

  1. The conclusion should be rewritten, as in the current form, it is only a general statement.

A: the conclusion is now revised from line 656-661.

Reviewer 3 Report

see document attached.

Author Response

Line/page

Reads

Should read / comment

26

and

and

A: now corrected

48

Culture moreover, revealed 

??

A: now corrected

198

2.8.16. S rRNA sequence

2.8. 16S rRNA sequence

A. now corrected

213

2.7. Statistical analysis of performance data

All statistical analysis should be in one only point, which could be sub-divided if necessary. 

A. now corrected according to the reviwer suggestion

301 and  lines 523-525

3.2. Enumeration of bacteria

I have nothing against traditional plaque enumeration of bacteria. However, since DNA was obtained, why no q-PCR analysis was done for specific relevant groups? For specific groups, quantitative analysis is more precise and reliable than sequencing.

A: Indeed qPCR is feasible but we are in the process of setting up the method in our lab and we will use it in the future when all primers are evaluated. Unfortunately this grant cannot cover the future work.

Also, as an attenuated live vaccine was used, why coccidia were not enumerated/quantified? 

A:nfortunately we did not prioritise this due to limited resources and we did not hypothesize that the major findings would arise from this analysis

423-432

margins

Now corrected and rephrased   431-434.

542

no substantial influence neither on broiler performance nor

no substantial influence either on broiler performance or

A.      Now corrected

Figures

Quite difficult to read due to letter size and lack of definition.

Now corrected

Round 2

Reviewer 2 Report

The authors adequately responded to all comments and performed all required modifications.

Author Response

.

Reviewer 3 Report

The manuscript animals-1475944- peer-review-v2 entitled “Impact of Dietary Sodium Butyrate and Salinomycin on Performance and Intestinal Microbiota in a Broiler Gut Leakage Model” is a revised version of a previous one dealing with comparisons between salinomycin and sodium butyrate as modulators of the intestinal microbiota composition in broilers. While authors addressed most of the queries raised in the previous revision, there are still some points which I think should be clarified: 1. There was no response on the issue that a control group with no challenge/additive should have been kept (see above). 2. The statistical analysis was not unified in one only point?

Author Response

While authors addressed most of the queries raised in the previous revision, there are still some points which I think should be clarified:

1.There was no response on the issue that a control group with no challenge/additive should have been kept (see above).

A: We agree with the reviewer that a negative control group would have been useful when evaluating the animal challenge model. However, before initiating this trial, we did a small pilot experiment, and compared to a non-challenged control group, we observed a significant decrease in bird weight gain related to the challenge. An even better design would have been a complete factorial design  including  three additional non-challenged groups, one receiving no additive and the others receiving butyrate and salinomycin, respectively. The housing of non-challenged and challenged birds in the same room is somewhat problematic, as cross contamination can occur. Keeping the non-challenged control birds in another room would have introduced a room effect as bias. In our view, in order to evaluate the effect of the feed additives, the inclusion of three challenged groups (as done in the present study) should be sufficient.

  1. The statistical analysis was not unified in one only point?

A: We have done all statistics in one point with sub sections in the materials and methods part of the manuscript.